# Growth and Phenology of *Vulpia Myuros* in Comparison with *Apera Spica-Venti*, *Alopecurus Myosuroides* and *Lolium Multiflorum* in Monoculture and in Winter Wheat

**DOI:** 10.3390/plants9111495

**Published:** 2020-11-05

**Authors:** Muhammad Javaid Akhter, Bo Melander, Solvejg Kopp Mathiassen, Rodrigo Labouriau, Svend Vendelbo Nielsen, Per Kudsk

**Affiliations:** 1Department of Agroecology, Research Centre Flakkebjerg, Aarhus University, DK-4200 Slagelse, Denmark; mohammadjaved@agro.au.dk (M.J.A.); bo.melander@agro.au.dk (B.M.); sma@agro.au.dk (S.K.M.); 2Department of Mathematics, Aarhus University, 8000 Aarhus C, Denmark; rodrigo.labouriau@math.au.dk (R.L.); svn@math.au.dk (S.V.N.)

**Keywords:** seed shedding, competitive ability, emergence, fecundity, target-neighborhood design, integrated weed management

## Abstract

*Vulpia myuros* has become an increasing weed problem in winter cereals in Northern Europe. However, the information about *V. myuros* and its behavior as an arable weed is limited. Field and greenhouse experiments were conducted in 2017/18 and 2018/19, at the Department of Agroecology in Flakkebjerg, Denmark to investigate the emergence, phenological development and growth characteristics of *V. myuros* in monoculture and in mixture with winter wheat, in comparison to *Apera spica-venti*, *Alopecurus myosuroides* and *Lolium multiflorum*. *V. myuros* emerged earlier than *A. myosuroides* and *A. spica-venti* but later than *L. multiflorum.* Significant differences in phenological development were recorded among the species. Overall phenology of *V. myuros* was more similar to that of *L. multiflorum* than to *A. myosuroides* and *A. spica-venti*. *V. myuros* started seed shedding earlier *than A. spica-venti* and *L. multiflorum* but later than *A. myosuroides*. *V. myuros* was more sensitive to winter wheat competition in terms of biomass production and fecundity than the other species. Using a target-neighborhood design, responses of *V. myuros* and *A. spica-venti* to the increasing density of winter wheat were quantified. At early growth stages “BBCH 26–29”, *V. myuros* was suppressed less than *A. spica-venti* by winter wheat, while opposite responses were seen at later growth stages “BBCH 39–47” and “BBCH 81–90”. No significant differences in fecundity characteristics were observed between the two species in response to increasing winter wheat density. The information on the behavior of *V. myuros* gathered by the current study can support the development of effective integrated weed management strategies for *V. myuros*.

## 1. Introduction

Noninversion tillage practices have increased throughout Europe in order to preserve soil productivity, reduce labor and costs for fuel among other reasons. Noninversion tillage in combination with a high frequency of winter cereals in the crop rotation has, however, caused new grass weed problems such as *Vulpia myuros* (L) C.C. Gmel. *V. myuros* has become an increasing weed problem in winter cereals in Northern Europe [1]. In Denmark, *V. myuros* was first reported in red fescue in the 1990s, and since then the infested area with *V. myuros* has increased significantly [2]. A more recent survey of grass weeds with a special focus on *V. myuros* revealed the higher infestations of *V. myuros* in some parts of Denmark [3]. Within the last 5 years, severe infestations of *V. myuros* have been reported more frequently in winter cereals. *V. myuros* is also reported in other European countries, for example, in UK, Romania and France, but is still considered a minor weed problem in those countries compared to Denmark [1]. A recent study from Denmark reported that ca. 400 *V. myuros* plants/m^2^ can reduce wheat yield by up to 50%, but ranked *V. myuros* as the least competitive grass weed in winter wheat in comparison with three other important grass species namely *Apera spica-venti* (L) P. Beauv., *Alopecurus myosuroides* Huds and *Lolium multiflorum* Lam [4]. The competitive ability of a weed species mainly depends on time of emergence, plant density and growth and development characteristics [5]. Weed species vary widely in their competitiveness with the crop depending on growth behavior and phenological features such as length of life cycles, initial growth rates, leaf area, root architecture, plant height and reproductive strategies [4,5]. The substantial differences in competitiveness between *V. myuros*, *A. spica-venti*, *A. myosuroides* and *L. multiflorum* could be related to their distinct growth behavior and phenological expressions [6]. A better understanding of the underlying processes that determine differences in competitiveness among these grass species can help develop cost-effective management strategies [7].

Acetolactate synthase (ALS) and acetyl CoA carboxylase (ACCase) inhibitors are the most widely used herbicides to control grass weeds in winter cereals. The natural tolerance in *V. myuros* to most ALS and ACCase inhibitor herbicides is a serious concern to its successful management [8]. Due to unsatisfactory performance of foliar applied herbicides, farmers are dependent on residual herbicides for the control of *V. myuros* in winter cereals. However, low efficacy of residual herbicides like prosulfocarb and pendimethalin on *V. myuros* is often reported by farmers [1]. Fewer chemical control options mean that integrated weed management (IWM) strategies, built on preventive and cultural methods, need to be adopted in order to manage *V. myuros* [1]. However, in order to develop effective IWM, basic information about the behavior of *V. myuros* is essential. The information about *V. myuros* and its behavior as an arable weed under Northern European conditions is limited according to a recent review by Akhter et al. [1]. An analysis of growth rate and developmental characteristics permits behavioral analysis of weeds in relation to ecological factors, which provide a useful tool for scheduling weed management tactics [9,10]. The primary objective of this study was to assess the emergence, growth and phenology of *V. myuros* in comparison with *A. spica-venti*, *A. myosuroides* and *L. multiflorum* in order to understand the plant traits that classify and determine their competitiveness.

Information on the competitive ability of a weed species at different growth stages is essential for its effective management because it determines the timing and intensity of control needed [8,11]. However, no information is available documenting the interaction of *V. myuros* with crop at its different growth stages. The secondary objective of this study was, therefore, to study sensitivity of *V. myuros* to winter wheat competition at different weed growth stages.

## 2. Material and Methods

### 2.1. Field Studies

A total of 2 field experiments were conducted on sandy loam soil at Flakkebjerg, Denmark (55°18′ N 11°23′ E) in 2017/18 and 2018/19. The experimental site has a temperate coastal climate, characterized by mild winters and cool summers. The prevailing climatic conditions were very different across the 2 experiments. A very wet autumn and a dry and warm summer prevailed in the first experiment (October 2017–August 2018) while the weather conditions in the second experiment (September 2018 –August 2019) were closer to normal conditions (Figure 1).

The trials were laid out in a randomized complete block design with 3 blocks; net plot size was 2.5 × 10 m. The experiment had 9 treatments: the 4 grass species grown in monoculture and in mixture with winter wheat, and winter wheat alone. Seeds of the 4 species were collected from a winter wheat field near the experimental site in July 2017, seed samples were stored in paper bags at 4 °C in the dark until seeding in the field experiments in October 2017 and September 2018. Seeding rates of the grasses were adjusted based on the results of germination tests to achieve a seedling density of 75 plants m^−2^ both in monoculture and in mixture with winter wheat. Winter wheat (cv. Maribos) was sown at 12.5 cm row spacing on October 10 in 2017, and September 13 in 2018 and seed rates were adjusted to target a field stand of 350–400 plants m^−2^. Seeds of grass weed species were drilled at shallow soil depth (0.5 cm) later on the same day. To obtain enough volume of grass seeds for the drilling operations, grass seeds were mixed with dead seeds of *Poa pratensis.* Seeds of *Poa pratensis* were killed by placing them in the oven at 100 °C for 144 h. In 2017/18, sowing was late due to unusual wet autumn, while crop was sown on time in 2018/19. The field was fertilized in April, with 180 kg ha^−1^ nitrogen, 34 kg ha^−1^ phosphorus and 86 kg ha^−1^ potassium. Broadleaved weeds were controlled in April applying 382.5 g ha^−1^ bromoxynil (Buctril EC 225, 225 g L^−1^ bromoxynil, Bayer CropScience) in 2018, and 7.5 g ha^−1^ tribenuron-methyl (Trimmer, 500 g kg^−1^ tribenuron-methyl, ADAMA Northern Europe BV) in 2019. Visual observations showed no grass weeds other than those sown in the experimental plots, which otherwise might have influenced the assessments of sown species.

Grass seedlings were counted in April at the tillering stage of the grass weeds in 2 randomly placed 0.25 m^2^ quadrats per plot in order to determine the grass weed density that had established in spring. To study the cumulative emergence of grasses as a function to thermal time, seedling counts were made in 3 fixed, 0.25 m^2^, quadrats per plot in the 2017/18 experiment. In 2018/19, counts were only made in 2 fixed, 0.25 m^2^, quadrats per plot because the results of 2017/18 experiment showed very low variation. Counts were made twice per week in the first 2 weeks after sowing and then biweekly; in total 10 and 6 emergence counts were made in 2017/18 and 2018/19, respectively.

The times needed to reach specific developmental stages of the grasses and the winter wheat were made 17 and 24 times in 2017/18 and 2018/19 respectively, on 5 marked plants of each species per plot, using the BBCH growth scale from BBCH 00 to BBCH 81 [12]. Plant seedlings emerging on the same date were marked to record the BBCH growth stages. The same 5 marked plants were used to study the timing of seed shedding using the BBCH growth scale. An assessment of the time when seeds started to separate “BBCH 93” from the panicles in *V. myuros*, *A. spica-venti*, *L. multiflorum* and seed heads in *A. myosuroides* was also made. This assessment was based on repeated observations that were started after reaching BBCH 75 for each species. Sowing date was used as the reference point for estimating thermal time required to reach key developmental growth stages.

Weed growth analyses were done 6 times from tillering until maturity by destructive measurements of plant biomass in two 0.25 m^−2^ quadrates in each plot. At each sampling time, grasses were counted and cut at ground level. Plant material was dried in the oven at 80 °C for 72 h to obtain dry matter (DM) content. Biomass was expressed as dry matter per plant to account for any variation in germination.

Germination of *A. spica-venti* was low resulting in much lower densities compared to the other species. Due to small number of plants, the study on *A. spica-venti* was limited to phenological characteristics only, while for the other species enough plants were present for studying the growth, phenology and fecundity traits as well.

To estimate the potential seed production, 5 plants of each species were collected randomly when *V. myuros* and *L. multiflorum* panicles, and *A. myosuroides* heads, had fully developed. Panicle length and head length were measured from the collected plants and numbers of panicles and heads per plant were counted. The correlation between number of seeds per panicle and panicle length and number of seeds per head and head length was used to estimate the seed production per plant as described by Melander [13]. The influence of crop competition on the correlation between seed number per panicle and head length was nonsignificant when comparing with pure stands of the grasses. Therefore, seed production was estimated from correlations based on pooled data of around 30 panicles and heads collected from each species growing in the absence and presence of crop. The estimated R^2^ values of correlation for *V. myuros*, *L. multiflorum* and *A. myosuroides* were 0.81, 0.71 and 0.71, respectively, in 2017/18, and 0.72, 0.63 and 0.75, respectively, in 2018/19 (data not shown).

### 2.2. Greenhouse Study

The target-neighborhood design was used to compare the competitiveness of *V. myuros* and *A. spica-venti* in response to increasing densities of neighboring winter wheat [14]. Six wheat (cv Maribos) densities of 0, 48, 96, 192, 288 and 576 plants m^−2^ were established by planting 0, 2, 4, 8, 12 and 24 plants in pots (23 cm diameter × 30 cm height) filled with a potting mixture consisting of sandy loam soil, peat and sand (2:1:1). The experiment included 4 factors: 2 grass species, 6 wheat densities, 2 growth stages of winter wheat and 3 harvesting times. The treatments were arranged in a complete randomized design with 4 replicates per treatment, resulting in 288 pots.

A template made of plastic sheet was used to achieve uniform distances between target and neighbor plants, and among neighbor plants [15]. Uniform-sized seeds of *V. myuros* and *A. spica-venti* were pregerminated in Petri plates, and transplanted into small paper pots when the leaf coleoptile was 1 cm long. Following, 1 equally sized weed seedling at the 2-leaf stage was transplanted to the center of each pot when the winter wheat plants were either at the 2 or the 3-4 leaf stage. Additional plants of winter wheat were grown in plastic trays and transplanted at the 2-leaf stage where winter wheat did not germinate or germinated later. The pots were placed in an unheated greenhouse on tables with automatic watering system with natural day length and irradiance reflecting the outdoor conditions. In total, 2 experiments were conducted in parallel with the field experiments; 1 in 2017/18 and 1 in 2018/19.

Above-ground dry matter (DM) of *V. myuros* and *A. spica-venti* were measured at “BBCH 26–29” (1st harvest) and “BBCH 39–47” (2nd harvest) of the grasses by cutting plants at the soil surface. In total, 3 replicates per treatment were harvested at each stage. The samples were oven dried for 24 h at 80 °C and dry matter (DM) was recorded. The number of tillers per grass plant was counted and plant height was measured. DM of *V. myuros* and *A. spica-venti* was also recorded at “BBCH 81–90” (3rd harvest). The seed production of target plants were estimated as described by Melander [13]. The seed production was estimated based on selected counts. The seeds were counted from around 45 panicles, and then lengths of corresponding panicle were measured. These values were used to find a correlation, a function was developed from the correlation and seed production was estimated based on this function. The estimated R^2^ value of correlation was 0.81 and 0.78 for *V. myuros* and *A. spica-venti*, respectively, in 2017 and 0.86 and 0.82, respectively, in 2018.

### 2.3. Data Analysis

Data from field and greenhouse studies were analyzed year-wise to account for the differences in climatic conditions between 2017/18 and 2018/19 growing seasons.

#### 2.3.1. Field Studies

Cumulative emergence of grasses were analyzed using 2-parameter log-logistic Equation (1) as a function of thermal time according to the time to event approach [16,17]:(1)$$E\left(t\right)=\frac{1}{1+exp\left[b\left(log\left(t\right)-log\left(GERM50\right)\right)\right]}$$
where *E* is the percent cumulative emergence at particular thermal time (*t*), *GERM50* is the thermal time to attain 50% emergence, and *b* is the emergence rate. The thermal time required for initiation of emergence (10% emergence = *GERM10*), 50% emergence (*GERM50*) and end of emergence (90% emergence= *GERM90*) were estimated with 95% confidence interval.

In total, 3 parameter Weibull Equation (2) and log-logistic functions Equation (3) were fitted to biomass data from 2017/18 and 2018/19, respectively, as a function of thermal time [17]:
(2)Y=(d)exp{−exp[c(log(t)−log(TIME50))]}
(3)$$Y=\frac{d}{1+exp\left[c\left(log\left(t\right)-log\left(TIME50\right)\right)\right]}$$

In Equations (2) and (3), *Y* is biomass accumulation of grasses; *c* is the rate of biomass production and *d* is the upper limit indicating total plant biomass. In the log-logistic model, *TIME50* is the thermal time (°C) needed to produce 50% of biomass, while the relation is more indirect for the Weibull function, where the *TIME50* value was calculated. Due to different climatic conditions across the 2 growing seasons, different models were fitted to the data from the experiments performed in 2017/18 and 2018/19. The model was checked with lack of fit test (*p* > 0.05), which showed Equations (2) and (3) could be used to describe the biomass accumulation of the grass species [18]. The sensitivity of grass weed species to winter wheat competition was compared using ratio of total biomass production *d* between grass species grew in the presence and absence of winter wheat competition. The parameter estimates were compared using a post hoc-t test.

The time to reach 6 developmental stages of grass weeds and crop from “BBCH 00” (sowing) to “BBCH 13” (3rd leaf enfolded), “BBCH 21” (1st side shoot visible), “BBCH 31” (beginning of elongation), “BBCH 51” (beginning of earing), “BBCH 81” (beginning of seed ripening) and “BBCH 93” (seed shedding) was estimated based on the thermal time scale. The statistical analyses were done separately for each growth stage. We compared the mean thermal times between species using a Monte Carlo permutation test (1000 random permutations). The growth stages were recorded at fixed time points, so the exact time when any plant reached a particular stage is unknown. We only had information on the intervals in which specific growth stages were reached. A standard procedure would be to select a point within each interval and use those as if they were the exact times but because each interval covered a range of growth stages, conclusions would depend heavily on the selected points. Instead, to avoid having to select points, we integrated out all possible exact times using Monte Carlo techniques. In each Monte Carlo iteration of the permutation test, we simulated an exact time for every observation assuming that every time point was uniformly distributed in the corresponding time interval. The variability of the estimates of thermal times was characterized by confidence interval (95% coverage) obtained using non-parametric bootstraps. The data on seed production was analyzed using a permutation test (with 1000 random permutations) combined with nonparametric bootstrap-based confidence intervals (95% coverage, 10,000 bootstrap samples). The suppression in seed production in grasses due to winter wheat competition was demonstrated calculating ratios of grass’s seed production in the presence and absence of winter wheat. The R-package postHoc [19] was used to perform post-hoc analyses involving the permutation tests.

The thermal time (°C) needed to estimate cumulative emergence of species was calculated using soil temperature, while thermal time required to estimate the BBCH growth stages and biomass production was calculated using air temperature. The base temperature was set to 1 °C for both grasses and winter wheat Equation (4) similar to previous studies [20,21,22,23]. If the mean temperature was at or below the base temperature, then thermal time values were assumed to be zero.
(4)$$\mathrm{Thermal}\;\mathrm{time}\;\left({}^{\circ}C\right)=\sum \left\{\left[\frac{\mathrm{Maximum}\;\mathrm{Daily}\;\mathrm{Temperature}+\mathrm{Minimum}\;\mathrm{Daily}\;\mathrm{temperature}}{2}\right]-1\right\}$$

#### 2.3.2. Greenhouse Study

A 2 parametric nonlinear hyperbolic model was used to analyze the data from the target neighborhood experiment [17]:
(5)$$Y=\frac{a}{1+\left[x/\mathrm{DENS}50\right]}$$
where Y is response variable (dry biomass, and potential seed production), *x* denotes wheat density (plant m^−2^), *a* is the response of target plant growing alone, DENS50 is the effective density of winter wheat reducing response of target plant by 50%. The competitiveness of 2 grass weeds and 2 winter wheat growth stages were compared in terms of the parameter DENS50 by means of a post-hoc test.

All the analyses were performed using the software R (R Foundation for Statistical Computing, Vienna, Austria, http://R-project.org). Time to event and dose response analysis were performed with R package ‘drc’ [18].

## 3. Results

### 3.1. Field Studies

#### 3.1.1. Weed Density

On average, plant population densities of *V. myuros*, *L. multiflorum*, *A. myosuroides* and *A. spica-venti* in the absence and presence of the winter wheat were 57, 68, 39, 10 and 44, 59, 19, 11 plants m^−2^, respectively, in 2017/18 and 64, 85, 57, 19 and 60, 73, 47, 16 plants m^−2^, respectively, in 2018/19. With one exception, the densities of grasses growing in the absence and presence of winter wheat were not significantly different (*p* > 0.05), therefore growth curves and fecundity characteristics were compared within grass species growing in the absence and presence of crop. In 2017/18, the density of *A. myosuroides* was significantly lower in the presence of the winter wheat than with no winter wheat.

#### 3.1.2. Cumulative Emergence

The cumulative emergence of the grasses in 2017/18 and 2018/19 as a function of thermal time are shown in Figure 2 and regression parameter estimates according to Equation (1) are presented in the Table 1. There were no significant differences in emergence whether grasses emerged alone or in a crop, therefore, regression estimates were compared among grasses ignoring the presence of the winter wheat (Table 1; Figure 2).

In 2017/18, emergence of *V. myuros* and *L. multiflorum* initiated (GERM10) at the same time (127 and 122 °C, respectively) (Table 1). The GERM10 values for *A. myosuroides* and *A. spica-venti* were significantly greater than for *V. myuros*. *L. multiflorum* was the first species to reach GERM50 (179 °C) and GERM90 (263 °C). *V. myuros* and *A. myosuroides* needed significantly longer thermal time than *L. multiflorum* to reach GERM50 (197 and 221 °C) and GERM90 (305 and 340 °C). The last species to reach GERM50 (265 °C) and GERM90 (418 °C) was *A. spica-venti*. In 2018/19, the GERM10 estimates for *V. myuros*, *L. multiflorum* and *A. myosuroides* were not statistically different. Similar to the previous year, *A. spica-venti* emergence occurred significantly later than the other species. In 2018/19, the overall emergence patterns were very similar to 2017/18, with the same relative differences observed among the grasses (Table 1), though the difference in GERM50 between *V. myuros* and *A. myosuroides* was nonsignificant.

The emergence rate parameters (*b*) indicated that *V. myuros* emerged at a similar rate as the other grass weeds. The only exception was in 2018/19, where *L. multiflorum* emerged significantly more rapidly than the other grass species.

#### 3.1.3. Phenological Behavior

There were no significant differences in the phenological development of the grass species or winter wheat grown in monoculture and in mixture with each other (*p* > 0.05). Hence, estimation of thermal time for attaining specific stages of the grasses and winter wheat were made on pooled data (Figure 3; Appendix A). Winter wheat needed significantly shorter thermal time to reach “BBCH 13” than the four grasses, for which no differences were found in both growing seasons. Mean thermal time required to attain “BBCH 21” was similar for winter wheat and grasses in 2017/18. However, in 2018/19, winter wheat reached “BBCH 21” first followed by *V. myuros*, *A. myosuroides* and *L. multiflorum,* among which no significant differences were observed. The thermal time for *A. spica-venti* to attain “BBCH 21” was longer than for the other species in 2018/19. *A. myosuroides* required the shortest thermal time while *A. spica-venti* required the longest thermal time to attain “BBCH 31”, “BBCH 50” and “BBCH 81” in both growing seasons. *V. myuros* attained the “BBCH 31” later than *L. multiflorum* in both growing seasons and later than winter wheat in 2017/18. There was no significant difference between *V. myuros* and winter wheat in reaching “BBCH 31” in 2018/19. Thermal time required to achieve “BBCH 51” and “BBCH 81” for *V. myuros* and *L. multiflorum* was significantly lower than for winter wheat. The thermal time for *V. myuros* to attain “BBCH 51” was significantly longer than for *L. multiflorum* in 2017/18 while no difference was observed in 2018/19. No statistical difference in thermal time to reach “BBCH 81” was found between *V. myuros* and *L. multiflorum* in 2017/18 and 2018/19.

#### 3.1.4. Biomass Accumulation

The biomass accumulation of the grasses in 2017/18 and 2018/19 as a function of thermal time are shown in Figure 4 and regression parameter estimates according to Equation (2) and (3) are presented in the Table 2 and Table 3. There were no significant differences in the rate of biomass accumulation (*c* parameter) and time to accumulate 50% biomass (*TIME50* parameter) between the grasses whether they grew in the absence or presence of winter wheat (Figure 4; Table 2 and Table 3), while significant differences were seen in total biomass production (*d* parameter). The ratio of total biomass production in the presence and absence of winter wheat varied among the grasses and years. The ratios in total biomass production of *V. myuros* grown in monoculture and with winter wheat were 0.11 and 0.18 in 2017/18 and 2018/19, respectively. The corresponding values for *A. myosuroides* and *L. multiflorum* were 0.29 and 0.42 in 2017/18, and 0.32 and 0.68 in 2018/19, respectively. On average, winter wheat suppressed the biomass of *V. myuros* more than of *A. myosuroides* and *L. multiflorum.*

#### 3.1.5. Seed Shedding and Seed Production

The grass species started seed shedding “BBCH 93” at the same thermal time whether they grew in monoculture or in mixture with a winter wheat crop (*p* > 0.05) (Figure 3; Appendix A). Hence, estimation of time for reaching seed shedding of grasses included data from all plots. Mature seeds of *V. myuros*, *A. spica-venti*, *A. myosuroides* and *L. multiflorum* started shedding at 1567, 1690, 1387, 1646 and 2165, 2249, 1870, 2213 thermal time (°C) corresponding to 30 June, 9 July, 19 June, 6 July in 2017/18 and 15 July, 19 July, 25 June, 18 July in 2018/19, respectively. Significant differences were observed among grasses (*p* < 0.001). For instance, *A. myosuroides* started shedding earlier than the other weeds in both years. *V. myuros* started shedding earlier than *L. multiflorum* and *A. spica-venti,* though the difference in relation to *L. multiflorum* was nonsignificant in 2017/18.

On average, 15,579, 3314 and 8508 seeds plant^−1^ were produced by *V. myuros*, *L. multiflorum* and *A. myosuroides*, in the absence of winter wheat. The presence of winter wheat significantly reduced (*p* < 0.001) seed production in grass species. *V. myuros* seed production was reduced to a greater extent by the presence of a winter wheat crop than *L. multiflorum* and *A. myosuroides,* which responded similarly to crop competition (Table 4). Seed production ratios of grasses grown in the presence and absence of crop competition were significantly lower for *V. myuros* (0.04 and 0.11 in 2017/18 and 2018/19, respectively) compared to *L. multiflorum* (0.20 and 0.22 in 2017/18 and 2018/19, respectively) and *A. myosuroides* (0.19 and 0.24 in 2017/18 and 2018/19, respectively) (Table 4).

### 3.2. Greenhouse Study

The dry matter (DM) and seed production of *V. myuros* and *A. spica-venti* decreased significantly with increasing winter wheat density (Appendix A). Similar results were observed at both winter wheat growth stages in both years. The average height of *V. myuros* and *A. spica-venti* plants were 74 cm and 77 cm when growing alone, and 117 cm and 129 cm when growing at the higher wheat density, respectively. Winter wheat densities above zero had no significant influence on the height of the two grass species.

The effective density of winter wheat (DENS50) reducing the biomass of the target plant by 50% significantly differed between *V. myuros* and *A. spica–venti* for the 2-leaf growth stage of winter wheat (Table 5). The greater estimated DENS50 values, the more competitive the weed plant was. At the first harvest “BBCH 26–29”, a higher DENS50 value was estimated for *V. myuros* than for *A. spica-venti* indicating that *V. myuros* was the most competitive of the two species. Conversely, at the second “BBCH 39–47” and third “BBCH 81–90” harvest times, higher DENS50 values were estimated for *A. spica-venti* than for *V. myuros*. This response was observed for the 2-leaf stage of winter wheat in both years. The same relative differences were observed for the 3–4 leaf stages of winter wheat; however, these differences were non-significant.

Generally, the estimated DENS50 values were lower when weeds were transplanted at the 3–4 leaf stage of winter wheat than the 2-leaf stage i.e., the more developed winter wheat plants were more competitive. However, the significant differences between growth stages were inconsistent across harvest times and years for each species. For instance, in 2017/18, the differences between growth stages were only significant for *V. myuros* at the first and third harvest time and for *A. spica-venti* only at the third harvest time. In 2018/19, significant differences were only observed at the third harvest time “BBCH 81–90” and only for *A. spica-venti*.

At maturity, no significant differences were observed in seed production between *V. myuros* and *A. spica-venti* grown in competition with winter wheat. Furthermore, a significant difference in seed production between the two winter wheat growth stages were seen in *V. myuros*, but not in *A. spica-venti*.

## 4. Discussion

### 4.1. Cumulative Emergence

Our results of *V. myuros* emerging at a similar rate as the other grass weeds are in line with those of Scherner et al. [23,24], who also found no significant differences in the emergence rate (*b*) between *V. myuros* and *A. spica-venti*. Regardless of similar emergence rates, the periodicity of emergence differed between *V. myuros* and *A. spica-venti.* For instance, *A. spica-venti* initiated emergence (GERM10) later and may therefore require more thermal time to complete the duration of emergence than *V. myuros*. The rapid emergence of *V. myuros* reported here presents an opportunity to control this species though stale or false seed-bed preparation. Preparing a seed-bed but postponing sowing means that many weed seedlings will emerge before crop sowing and can be controlled either chemically or mechanically prior to sowing. A recent study showed that a false seed-bed could reduce the infestation of *V. myuros* by up to 80% [25].

Weed and crop emergence time determine the interaction between weed and crop growth [10]. The faster a species emerges, the more it grows at the expense of its competitor [10]. The rapid emergence rate of *V. myuros* indicates the higher early season competitiveness of *V. myuros* compared to *A. spica-venti*. Akhter et al. [4] demonstrated the potential of *V. myuros*’ strong competitive ability with winter wheat early in the growing season. The results highlight the significance of controlling this species early in the growing season.

Under natural conditions, grass weed species have distinct emergence patterns [23]. To include the effect of emergence on weed competitiveness, thermal time required to reach key BBCH stage was calculated starting on the day of sowing. Germination pattern of dry and newly harvested seeds may deviate from the seeds in the natural soil seedbank. Nondormant seeds in the soil seedbank are already imbibed and ready to germinate immediately after ploughing. This may influence the time of initiation of germination and emergence. Field trials were initiated approximately 4 and 16 months after seed collection in 2017/18 and 2018/19 and seed storage duration and conditions were expected to break primary seed dormancy [24]. Moreover, the relative difference in emergence duration between the species were consistent across years. In addition, the emergence of the four species included in the present study is in accordance with previous studies conducted with a natural soil seed bank under normal growing conditions [23,26,27,28]. Hence, it can be assumed that seed dormancy did not influence the germination or germination rate in the present study [21,23,24,25].

### 4.2. Growth and Phenology

Winter wheat, in general, was faster to reach “BBCH 13” and “BBCH 21” than the grass species, which is in accordance with previous findings for *A. myosuroides* [29]. Sowing of winter wheat later than the optimum time exposed the early plant growth stages to reduced light and temperature conditions that resulted in slower growth and reduced tillering [30,31,32,33], which could partially explain the non-significant difference in thermal time among plant species to reach “BBCH 21” in 2017/18 (Figure 3). The information of thermal time needed for leaf and tiller appearance could help optimize herbicide application timings and dosages for effective control [20]. Timing of herbicide application is critical because younger weed plants are often more sensitive than more-developed stages [11,34]. This study suggest that early post emergence control of *V. myuros* might be effective up to tillering stage “BBCH 21” of winter wheat because at this stage *V. myuros* was still at the leaf development stage “BBCH 13” [11]. The study showed that stem elongation “BBCH 31” started earlier in *A. myosuroides* than in the other grass weed species and wheat. This finding is similar to that by Chauvel et al. [20]. The slower development of *A. spica-venti* started at BBCH 21, and is in accordance with results reported by Soukup et al. [35]. *V. myuros* and *L. multiflorum* reached the advanced growth stages “BBCH 51” and “BBCH 80” simultaneously and slower than *A. myosuroides* but faster than *A. spica-venti*. Overall, the life cycle of *V. myuros* was more similar to winter wheat and *L. multiflorum* than to *A. myosuroides*, which completed its life cycle much faster than winter wheat.

The greater reduction in biomass of *V. myuros* than *L. multiflorum* and *A. myosuroides* reflects a higher sensitivity of *V. myuros* to winter wheat competition despite the potential of *V. myuros* to produce more biomass when growing in a pure stands Table 3. The differences in competitiveness between *V. myuros* and *L. multiflorum* seems not to be related to their developmental rates but the potential of resource utilization and plant size attributes. *V. myuros*, probably due to its short stature, shallow root system [36] and low specific area, is less competitive. In contrast, *L. multiflorum*, due to its deeper roots, higher nutrient uptake, greater stature and high specific leaf area, is highly competitive [37,38]. Weed species with slower developmental rates and longer life cycles are more resource demanding and therefore more competitive, while species with a fast development and short life cycle have shorter time for extraction of resources and, thus, less competitive [9,39]. Due to a slower development, *A. spica-venti* tends to stay green longer having a higher ability to absorb nutrients from the soil [40]. Because of a short life cycle, *A. myosuroides* tends to compete with the crop for a shorter duration than *A. spica-venti*, especially at later crop growth stages [13]. The results from the current study do not allow us to establish any general relationship between weed competitiveness and specific plant traits [7], but the differences can be related to species specific growth and development traits, and periods of active growth [4].

### 4.3. Seed Shedding and Seed Production

*V. myuros* started seed shedding later than *A. myosuroides* but earlier than *A. spica-venti* (Figure 3). The pattern of seed shedding between *V. myuros* and *L. multiflorum* was inconsistent across the years probably because of different growing conditions between the 2 years (Figure 1). Similar to our results, Bitarafan and Andreasen [41] reported earlier seed shedding in *A. myosuroides* than *A. spica-venti* over two growing seasons, and variation in the timing of seed shedding due to different climatic conditions was observed between the years. The percentages of seeds retained on the plants of *A. spica-venti* and *A. myosuroides* were 35% and 34%, respectively, at winter wheat harvest in Denmark [41]. Walsh et al. [42] reported that *L. multiflorum* retained up to 58% of the seeds at harvest in US. If seeds retain on the plants, they can be harvested and destroyed which could potentially reduce the soil seedbank markedly over time. The percentages of seed retained on the plants was not measured in the present study. Visual observations showed that a significant proportion of seeds were retained on *V. myuros* at the time of wheat harvest, but the ears were not upright like *A. myosuroides*, *L. multiflorum* and *A. spica-venti* but bending downwards making it less likely that the retained seeds will be collected by the combiner at harvest. There is no published literature reporting the percentage of seeds retained on the *V. myuros* plant at the time of harvest.

This study explored seed production of four grass weeds growing in monoculture and in competition with winter wheat (Table 4). The presence of a crop canopy can reduce the seed production of weeds by influencing their growth and development [15]. The higher suppression in seed production in *V. myuros* compared to *L. multiflorum* and *A. myosuroides* suggests a potential of suppressing *V. myuros* by growing competitive crops, which should be considered as part of an IWM strategy.

### 4.4. Greenhouse Study

The differences in competitiveness of *V. myuros* and *A. spica-venti* recorded in the target-neighborhood experiment can be ascribed to their developmental characteristics as revealed in the field experiments. Faster early growth of *V. myuros* than *A. spica-venti* makes it a stronger competitor at early growth stages, however, opposite effects were observed at later growth stages. Similar to our results, Dillon and Forcella [10] also noticed a rapid initial growth rate of *V. myuros*. Although *A. spica-venti* is less competitive during the early growth stages, its growth rate significantly increases in spring and it can outgrow a crop before heading stage, thus, compete more vigorously with crop at later stages [13,43]. The competitive differences of *V. myuros* between early and late growth stages could be due to a more pronounced response to light than *A. spica-venti*, i.e., it is more competitive early in the season where shading by the crop is lower than later in the season when crop canopy is dense. Tozer [36] found that *V. myuros* was a poor competitor for light when growing at high crop densities. *V. myuros* does not grow as high as *A. spica-venti* and has less specific leaf area [36], which may explain why *V. myuros* is less competitive than *A. spica-venti*. It was not possible to measure seed production, but the indirect estimation of seed production was a good tool for measuring fecundity traits of grass species [15]. The winter wheat density had less effect on the biomass of *A. spica-venti* than *V. myuros* at maturity; however, the differences in seed production were nonsignificant between the two species. This finding revealed the potential of *V. myuros* to produce enough seeds even with lower accumulated biomass relative to *A. spica-venti*. Nonsignificant differences in numbers of tillers between *V. myuros* and *A. spica-venti* at maturity is the possible explanation of the nonsignificant difference in the per plant seed production to increasing winter wheat density between the two species. On average, the number of tillers per plant for *V. myuros* and *A. spica-venti* at maturity were 22 and 13 growing alone, and 3 and 3 growing at the higher wheat density, respectively. Similarly, to our results Menchari et al. [44] also found a correlation between number of tillers and the total seed production in *A. myosuroides*.

The results of this study have a direct relevance for understanding not only the features that determine the weedy characteristics of *V. myuros* but also for devising solutions. Due to natural tolerance to most widely used herbicides, chemical control of *V. myuros* is not highly effective [1]. Hence, farmers are reliant on IWM for reducing the impact of *V. myuros* on crop yield. Information on emergence, life cycle, initial growth rates, patterns of biomass accumulations and seed production potential will help to understand the behavior of *V. myuros* in comparison to the more common grass weed species, *A. spica-venti*, *A. myosuroides* and *L. multiflorum.* The results from this study will provide a scientific basis for developing IWM strategies for *V. myuros*. Based on this study, *V. myuros* is classified as less competitive to winter wheat in terms of biomass production and fecundity than the other studied grass weed species. Moreover, our results showed that delayed sowing is an important cultural practice, which provides a window to control the rapid emerging early cohorts of *V. myuros* mechanically or chemically before sowing. A recent study confirmed the impact of crop competition and delayed sowing in limiting the growth and seed production of *V. myuros* [4]. *Vulpia myuros* exhibits phenological traits of a winter annual with a life cycle similar to that of winter wheat, which suggests that inclusion of spring sown in the crop rotation should be effective managing this species. Recently, Scherner et al. [45] have found that the inclusion of spring-sown crops in a 12-year crop rotation dominated by winter cereals significantly suppressed weed seed bank of *V. myuros,* and this effect was more pronounced for *V. myuros* than *A. spica-venti*. Seed biology studies have revealed that *V. myuros* has lower seed longevity in the soil than other grass species, which is the main reason that infestation of *V. myuros* is mainly associated with noninversion tillage practices [10,46,47]. Even though soil cultivation is an effective measure to manage *V. myuros*, no-till is being widely adopted in order to preserve soil productivity. Strategic ploughing, also referred to as occasional ploughing, could be another solution to prevent the build-up of large infestations of *V. myuros* in no-till systems [1].

Weed population dynamic models can assist in the practical implementation of IWM by predicting long-term effects of different weed management tactics on weed population dynamics [48], and results from this study can be employed to parameterize weed population dynamic models in order to predict long-term effects of different control measures on *V. myuros.*

## 5. Conclusions

The results from this study showed that *V. myuros* can emerge fast and exert significant early competition on the crop. The phenology of *V. myuros* was more similar to that of *L. multiflorum* and winter wheat than that of *A. myosuroides* and *A. spica-venti*. Higher sensitivity of *V. myuros* to winter wheat competition in terms of biomass and fecundity suppression suggest that *V. myuros* is the least competitive of the studied grass weed species. The basic information produced in this study adds to the understanding of *V. myuros* behavior as an arable weed. This helps to the formulation and optimization of IWM strategies for the control of *V. myuros*.

## Figures and Tables

**Figure 1 plants-09-01495-f001:**
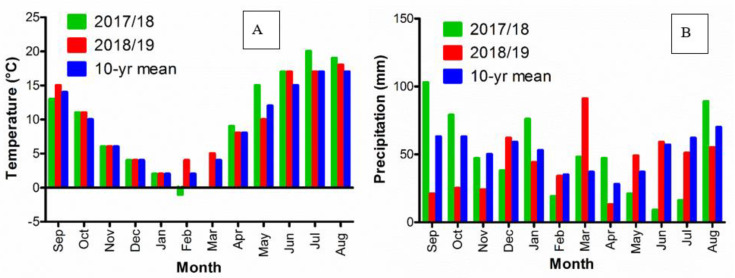
Weather data for 2017/18, 2018/19, and 10 yr means from 2011 to 2020. (**A**) Monthly average air temperature and (**B**) precipitation.

**Figure 2 plants-09-01495-f002:**
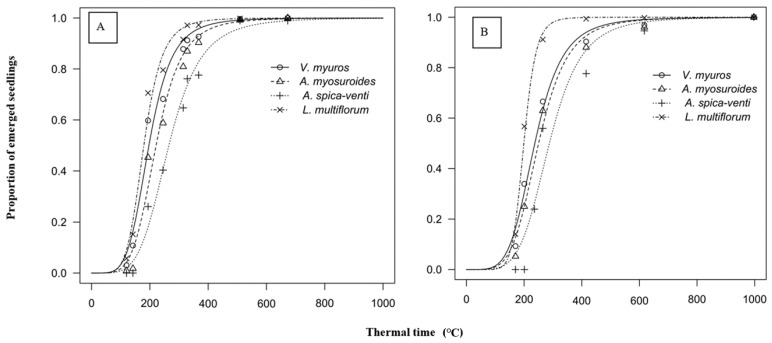
Cumulative emergence dynamics of grass species as a function of thermal time (°C) in 2017/18 (**A**) and 2018/19 (**B**). Regression equation and parameter estimates given in Table 1.

**Figure 3 plants-09-01495-f003:**
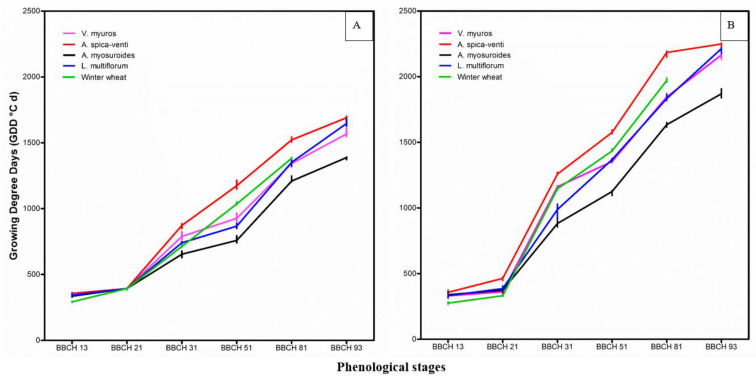
Thermal time (°C) for the grass weeds and winter wheat to reach various phenological stages in 2017/18 (**A**) and 2018/19 (**B**). Mean values and 95% bootstrap confidence intervals are presented in Appendix A (Appendix A (2017/18) and Appendix A (2018/19)).

**Figure 4 plants-09-01495-f004:**
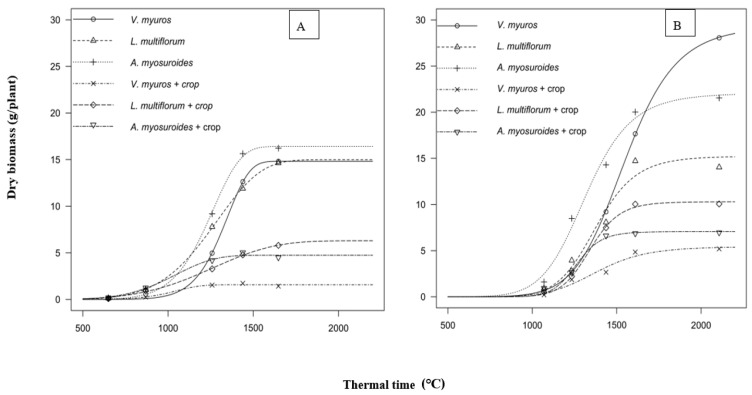
Per plant biomass accumulation of *V. myuros*, *L. multiflorum* and *A. myosuroides* grown in the absence and presence of winter wheat crop in relation to thermal time (°C) in 2017/18 (**A**) and 2018/19 (**B**). Regression equation and parameter estimates given in Table 2 (2017/18) and Table 3 (2018/19).

**Table 1 plants-09-01495-t001:** Regression parameter estimates from log-logistic model Equation (1) for cumulative percent emergence of grass weeds in 2017/18 and 2018/19.

	Regression Parameters ^a^							
	*b* (95% CI) ^c^		GERM10 (°C) (95% CI)		GERM50 (°C) (95% CI)		GERM90 (°C) (95% CI)	
	2017/18	2018/19	2017/18	2018/19	2017/18	2018/19	2017/18	2018/19
*V. myuros*	−5.0 (−5.4, −4.6) (1)	−4.9 (−5.3, −4.5) (1)	127 (119–134) (1)	152 (142–161) (1)	197 (189–205) (1)	237 (227–247) (1)	305 (286–324) (1)	371 (347–395) (1)
*L. multiflorum*	−5.7 (−6.2, −5.2) (2)	−9.5 (−10.4, −8.5) (2)	122 (116–128) (2)	158 (153–163) (2)	179 (173–186) (2)	199 (195–203) (2)	263 (248–277) (2)	251 (242–260) (2)
*A. myosuroides*	−5.0 (−5.5, −4.7) (3)	−5.0 (-5.4, −4.5) (3)	144 (136–151) (3)	161 (152–170) (3)	221 (212–230) (3)	250 (240–260) (3)	340 (320–361) (3)	389 (364–414) (3)
*A. spica-venti*	−4.8 (−5.2, −4.4) (4)	−5.2 (−5.6, −4.7) (4)	168 (156–178) (4)	188 (178–198) (4)	265 (254–276) (4)	288 (276–299) (4)	418 (393–444) (4)	440 (412–468) (4)
Significance levels ^b^	1 vs. 2, *p* = 0.059	1 vs. 2, *p* < 0.001	1 vs. 2, *p* = 0.350	1 vs. 2, *p* = 0.260	1 vs. 2, *p* < 0.001	1 vs. 2, *p* < 0.001	1 vs. 2, *p* < 0.001	1 vs. 2, *p* < 0.001
	1 vs. 3, *p* = 0.778	1 vs. 3, *p* = 0.835	1 vs. 3, *p* < 0.001	1 vs. 3, *p* = 0.166	1 vs. 3, *p* < 0.001	1 vs. 3, *p* = 0.067	1 vs. 3, *p* = 0.017	1 vs. 3, *p* = 0.330
	1 vs. 4, *p* = 0.609	1 vs. 4, *p* = 0.488	1 vs. 4, *p* < 0.001	1 vs. 4, *p* < 0.001	1 vs. 4, *p* < 0.001	1 vs. 4, *p* < 0.001	1 vs. 4, *p* < 0.001	1 vs. 4, *p* < 0.001
	2 vs. 3, *p* = 0.104	2 vs. 3, *p* < 0.001	2 vs. 3, *p* < 0.001	2 vs. 3, *p* = 0.550	2 vs. 3, *p* < 0.001	2 vs. 3, *p* < 0.001	2 vs. 3, *p* < 0.001	2 vs. 3, *p* < 0.001
	2 vs. 4, *p* = 0.016	2 vs. 4, *p* < 0.001	2 vs. 4, *p* < 0.001	2 vs. 4, *p* < 0.001	2 vs. 4, *p* < 0.001	2 vs. 4, *p* < 0.001	2 vs. 4, *p* < 0.001	2 vs. 4, *p* < 0.001
	3 vs. 4, *p* = 0.424	3 vs. 4, *p* = 0.627	3 vs. 4, *p* < 0.001	3 vs. 4, *p* < 0.001	3 vs. 4, *p* < 0.001	3 vs. 4, *p* < 0.001	3 vs. 4, *p* < 0.001	3 vs. 4, *p* < 0.001

^a^ E(t) = 1/(1 + exp[*b*(log(t) − log(*GERM50*))]). E is the cumulative emergence, *GERM50* is the thermal time in needed to attain 50% emergence and *b* is the rate of emergence. ^b^ Parameter estimates were compared within a year by t-tests at the 5% level of significance. ^c^ Abbreviations: *b =* emergence rate; CI = confidence interval; GERM10, GERM50, and GERM90 = thermal time (°C) needed for 10%, 50% and 90% emergence, respectively.

**Table 2 plants-09-01495-t002:** Regression parameter estimates from weibull function Equation (2) fitted to per plant biomass accumulation of grass weeds in the absence and presence of winter wheat in 2017/18. Regression estimates were compared within grasses grown in the absence and presence of winter wheat at the 5% level of significance. Standard errors are given in parentheses.

	Regression Parameters ^a^								
	*c*			TIME50 (°C)			*d*		
Weed species	−Wheat ^c^		+Wheat	−Wheat		+Wheat	−Wheat	+Wheat	Ratio ^b^
*V. myuros*	11.4 (2.1)	ns	8.3 (31.2)	1317.0 (19.6)	ns	1020.0 (587.7)	14.8 (0.7)	1.5 (0.6)	0.11 (0.04) *p* < 0.001
*L. multiflorum*	6.0 (1.3)	ns	4.5 (2.3)	1251.0 (37.2)	ns	1235.0 (178.3)	14.9 (1.2)	6.2 (2.2)	0.42 (0.15) *p* < 0.001
*A. myosuriodes*	8.7 (2.3)	ns	5.6 (2.4)	1230.0 (23.5)	ns	1016.0 (107.7)	16.4 (0.7)	4.7 (0.5)	0.29 (0.04) *p* < 0.001

^a^Y=(d) exp{−exp[c(log(t)−log(TIME50))]} Y is biomass accumulation of grasses; c is rate of biomass production, TIME50 is the thermal time (°C) needed to produce 50% of biomass, *d* is upper limit indicating total plant biomass. ^b^ Estimated ratio in *d* value at the 5% level of significance within grass species that grew in the presence and absence of winter wheat. A lower ratio indicates a greater sensitivity of a species to produce biomass with winter wheat competition. ns, nonsignificant (nonsignificant difference within grass species when they grew in the absence and presence of winter wheat). *c* Abbreviation: −Wheat indicating when grasses grew in the absence of winter wheat; +Wheat indicating when grasses grew in the presence of winter wheat

**Table 3 plants-09-01495-t003:** Regression parameter estimates from log-logistic function fitted to per plant biomass accumulation of grass weeds in the absence and presence of winter wheat in 2018/19. Regression estimates were compared within grasses when they were grown in the absence and presence of crop at the 5% level of significance. Standard errors are given in the parenthesis.

	Regression Parameters ^a^								
	*c*			TIME50 (°C)			*d*		
Weed species	−Wheat ^c^		+Wheat	−Wheat		+Wheat	−Wheat	+Wheat	Ratio ^b^
*V. myuros*	−10.1 (1.7)	ns	−9.5 (5.9)	1548.0 (36.5)	ns	1384.0 (124.6)	29.3 (2.6)	5.4 (1.2)	0.18 (0.04) *p* < 0.001
*L. multiflorum*	−11.7 (2.6)	ns	−15.4 (5.5)	1380.0 (38.2)	ns	1343.0 (44.8)	15.2 (1.2)	10.3 (1.0)	0.68 (0.08) *p* < 0.001
*A. myosuroides*	−9.8 (1.7)	ns	−17.3 (14.4)	1319.0 (34.9)	ns	1263.0 (51.7)	22.0 (1.6)	7.1 (0.9)	0.32 (0.05) *p* < 0.001

^a^Y=d/(1+exp[c(log(t)−log(TIME50))]) Y is biomass accumulation of grasses, c is rate of biomass production, TIME50 is the thermal time (°C) needed to produce 50% of biomass, d is upper limit indicating total plant biomass ^b^. Estimated ratio in d value at the 5% level of significance within grass species that grew in the presence and absence of wheat. A lower ratio indicates a greater sensitivity of a species to produce biomass with winter wheat competition. ns, nonsignificant (nonsignificant difference within grass species when they grew in the absence and presence of wheat). c Abbreviation: −Wheat indicating when grasses grew in the absence of wheat; +Wheat indicating when grasses grew in the presence of wheat.

**Table 4 plants-09-01495-t004:** Mean per plant seed production and ratio of seed production of annual grass weeds in the presence and absence of winter wheat. For grass species with the same labels (c and d) indicate that the ratio did not differ significantly. 95% bootstrap confidence intervals are given in parentheses.

	2017/18			2018/19		
	−Wheat ^a^	+Wheat	ratio ^b^	−Wheat	+Wheat	Ratio
*V. myuros*	14478 (7833–22715)	574 (472–678)	0.04 (0.02–0.07) c	16680 (15019–18446)	1822 (1509–2169)	0.11 c (0.10–0.12)
*L. multiflorum*	2364 (2136–2724)	426 (339–818)	0.20 (0.16–0.25) d	4264 (3691–4774)	958 (752–1158)	0.22 d (0.20–0.24)
*A. myosuroides*	4896 (4320–5504)	891 (702–1097)	0.19 (0.14–0.24) d	12120 (10119–14143)	2912 (2172–3787)	0.24 d (0.21–0.27)

^a^ Abbreviation: −Wheat means that the grasses grew in the absence of winter wheat; +Wheat means that the grasses grew in the presence of winter wheat. ^b^ Ratio of seed production in the presence and absence of crop, lower ratio indicates the higher sensitivity of a species to produce seeds with crop competition.

**Table 5 plants-09-01495-t005:** The estimated effective density of winter wheat (DENS50) required to suppress 50% responses of *V. myuros* and *A. spica-venti* against increasing winter wheat densities at two different crop growth stages (1–2-leaf stage, 3–4-leaf stage) in 2017/18 and 2018/19 using a target-neighborhood design. Values in parentheses indicate standard errors. DENS50 (winter wheat plants m^−2^) parameter obtained by using two parametric non-linear hyperbolic equation ^a.^

2017/18 Experiment		1-2-Leaf Stage			3-4-Leaf Stage			
Plant traits	Harvest time	*V. myuros* (1)	Significance level ^b^	*A. spica-venti* (2)	*V.myuros* (3)	Significance level ^b^	*A. spica-venti* (4)	Significance level ^c^
Biomass	BBCH (26–29)	104 (21.0)	*p* = 0.008	35 (9.7)	51 (25.8)	ns	16 (7.2)	1 vs. 3, *p* = 0.035, 2 vs. 4 = ns
	BBCH (39–47)	28 (6.1)	*p* = 0.002	92 (31.0)	47 (39.3)	ns	52 (23.0)	1 vs. 3 = ns, 2 vs. 4 = ns
	BBCH (81–90)	30 (5.6)	*p* = 0.043	82 (24.4)	11 (3.52)	ns	24 (11.7)	1 vs. 3, *p* = 0.004, 2 vs. 4, *p* = 0.037
Potential seed production	BBCH (81–90)	39 (7.7)	ns	35 (8.9)	16 (7.39)	ns	20 (7.7)	1 vs. 3, *p* = 0.038, 2 vs. 4 = ns
**2018/19 experiment**								
Biomass	BBCH (26–29)	30 (4.0)	ns	17 (4.4)	24 (10.2)	ns	14 (12.7)	1 vs. 3 = ns, 2 vs. 4 = ns
	BBCH (39–47)	10 (4.5)	*p* = 0.0013	29 (6.8)	9 (9.4)	ns	14 (7.4)	1 vs. 3 = ns, 2 vs. 4 = ns
	BBCH (81–90)	14 (5.9)	*p* < 0.001	51 (9.2)	7 (3.4)	ns	14 (4.2)	1 vs. 3 = ns, 2 vs. 4, *p* < 0.001
Potential seed production	BBCH (81–90)	18 (4.6)	ns	43 (15.8)	6 (2.5)	ns	14 (7.8)	1 vs. 3, *p* = 0.018, 2 vs. 4 = ns

^a^Y=a/(1+[x/DENS50]) Y is response variable, *x* denotes wheat density (plant m^−2^), *a* is the response of target plant growing alone, DENS50 is the effective density of winter wheat reducing response of target plant by 50%. ^b^ Parameter estimates were compared between species by t-tests at the 5% level of significance. ^c^ Parameter estimates were compared between crop growth stages within target plants. ns, nonsignificant difference.

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
