# Peer review of "Growth and Phenology of Vulpia Myuros in Comparison with Apera Spica-Venti, Alopecurus Myosuroides and Lolium Multiflorum in Monoculture and in Winter Wheat"

_plants, 2020, doi:10.3390/plants9111495_

Round 1
Reviewer 1 Report
Vulpia myuros is a species that is relatively little studied and there is a need for more knowledge of this weed species. The present work consists of a relatively wide area (emergence, phenology, growth and seed production). It contains a relatively high number of experiments all conducted over two growing seasons and the results are presented in an efficient way. I think it is a well written paper and relatively easy to follow. I have however a few issues that could be improved.
Abstract: the emergence part of the study is not mentioned in the abstract.
Introduction: the introduction is short, but set the reader well into the background of the study. I question if some more references could be cited for each statement.
Materials and Methods:
Sometimes there is lack of information such as literature reference to the BBCH- scale at first mention (line 98) and the method in the greenhouse study on competition (the target- neighbourhood design, line 127).
The figures and tables need to be improved and checked for spelling errors and missing information: Fig. 1: part of Fig. 1B not visible, Fig. 2: Should include explanations/units on the y-axis. Table 2 & 3: explanation to footnote index ‘a’ missing. Differences in d values in absolute numbers are given in the last column. Would it not be transferable to practical situations to calculate it also as percentage of reduction? In Table 3 the footnotes are above the table, it should be below.
I think the unit for thermal time is ‘ËšC d’ and not only ‘ËšC’ as written several places e.g. in line 188, Fig. 2 and Fig. 4, Table 1. This should be checked in the text, tables and figures.
Sometimes only ‘time’ is written- I assume that the authors mean ‘thermal time’, since thermal time seam to used in the whole study (e.g Line 173, 174. 188). Please replace ‘time’ with ‘thermal time’.
What was drilling depth in cm of the grass seeds (line 86)? What kind of herbicide was used to control dicot weeds (line 89)?
Line 94-95: When does the counting of emerged plants started, was it in autumn or spring? You mention before that counting was done in April at tillering stage (line 90). Please state if the tillering stage is given for the crop or the grass weeds.
Line 130, is a comma missing between ‘soil’ and ‘peat’? What was the soil type used?
Line 134: what kind of template was used to achieve uniform distances?
Line 140-142: Day length, irradiance and irrigation mentioned, but nothing about temperature. What was the temperature in the greenhouse? Was it unheated?
Line 143-144: I assume that the BBCH mentioned is that of the crop (wheat). Or was it that of the grass weeds?
Line 147-148. It is stated that the seed production was estimated as described by Melander – were the seeds counted or was it estimated by function?
Results:
Line 217: Should ‘wheat’ be included after ‘winter’?
Line 217-218: the last part of the sentence seam to belong to the discussion part.
The experiments were conducted from autumn over winter to next year. I wonder when the winter occurred in the emergence and phenology studies - could it be indicated somehow in Fig. 2 and 3? This influence the possibilities to use the work as background for integrated weed management, e.g. different herbicides may be authorized in autumn and spring in winter wheat (see also line 396-398).
Line 331-332: Plant height was unaffected by winter wheat density and not any was presented. I think it could be of interest for competitive ability to indicate in the text how tall the plants were.
Discussion:
Line 360: Scherner et al. is mentioned twice.
Line 396-398: was the appearance of leaf or tillers in autumn or spring? This could influence possibilities for weed control
Line 417-418: It is mentioned that species with fast development and short life cycle are less competitive. I think that not always may be true. Relative emergence time of crop and weed can be of importance and early emergence could be a competitive advantage.
Line 443: ‘seed characteristics’ – it seam that the only characteristics presented is seed production, could only write ‘seed production’ instead of ‘seed characteristics’.
In the discussion section I think it should be discussed more how this study can help farmers in developing integrated weed management systems and how to use this study to parameterize weed population models.
Conclusion:
I think conclusions on emergence is lacking. New aspects not or little discussed earlier were introduced (parameterizing weed population models, some fragment of implications for weed management is mentioned in the discussion)- it should have been mentioned earlier in the paper.
Reference list:
Reference no. 2 is not published, I think it should not be listed in the References, only be mentioned in the text as Akhter et al. (unpublished).
Reference no. 18 and 23 seem to be the same, although there is an extra author is in no. 18.
Author Response
Dear Reviewer,
Thanks for the opportunity to submit a revised version of our manuscript. Our point-to-point response to the comments of the reviewers are in italic. In addition to the reviewer's comments and suggestions, thettt manuscript was revised according to the directions from the authors.

Reviewer 2 Report
Comments to the Authors
Introduction
Line 35 – “have” not “has”
Line 38 – suggest this sentence could be “A recent study from Denmark reported that 405 V. myuros plants/m2 reduced wheat yield by up to 50%.”
Line 43 – what is it about their growth behaviour and phenological expression that results in differences in competitiveness?
Line 46 – need to explain ACCOSE and ALS
Line 48 – place “(IWM)” after “management”
Line 60 – replace “different weed growth stages” by “weeds at different growth stages”
Line 61 – “varies” not “vary”
Materials and Methods
Line 71 – suggest replacing ‘was observed’ by “occurred”
Line 80 – storing seeds in paper bags at 4°C was unwise. Seed moisture content would have increased to come into equilibrium with the high moisture content of the air (relative humidity) resulting in a loss in germination.
Line 87 - 89 – was this only in 2017/18? What about the second year?
Line 101 – what does “start to loosening” mean?
Line 104 – “analyses” not “analysis”
Line 188 – thermal time?
Table 1 – I suggest this should be placed in the supplementary files
Figure 2 – what are the units for the vertical axis?
Line 244-245 – you did not measure germination in the field
Figure 2 – why include this Figure if the information it contains is not mentioned in the text?
Line 247 – better to say “there were no significant differences in phenological development” (delete “were observed”)
Line 250-251 – make it clear that this was the case in both years
Line 252 – if differences are not significant then winter wheat cannot be “first”
Line 255 – make it clear that this happened in both years
Table 2/3 – a Table must be able to “stand alone” which means it must be understandable without having to refer to the text. You will need to explain what c, TIMF50 and d are (or better still replace them with rate of biomass accumulation, time to accumulate 50% biomass and total biomass production.
Figure 4 – why is this included when you don’t refer to it in the results?
Lines 31-302 – how do Figure 3 and Tables S1 and S2 show that the grasses had started seed shedding?
Table 4 – the a and b are confusing – find other symbols for the footnote
Line 348 – ‘no significant differences’ not ‘non-significant’.
Discussion
Line 360 – same comment as for Line 348
Line 363 – you stated that A. spica-venti had poor germination, indicating a physiological deteriorating seed lot (possible accelerated by the storage method). Could this have been a factor in the slower emergence?
Line 365-366 – are you confident about this statement? Why do small seeds require a longer time to germinate than big seeds? Seed size is rarely a factor in germination responses.
Line 368 – cold storage alone does not break primary dormancy. Time is more likely the reason the dormancy was broken.
Conclusion
Line 471 – what strategies might be possible?
General – the text has many errors, particularly mismatching of tense. I suggest a careful read through to correct language mistakes.
References – style differences throughout; correct for consistency
Author Response
Dear Reviewer,
Thanks for the opportunity to submit a revised version of our manuscript. Our point-to-point response to the comments of the reviewers are in italic. In addition to the reviewer's comments and suggestions, the manuscript was revised according to the directions from the authors.
